# The role and risks of selective adaptation in extreme coral habitats

**Federica Scucchia** ®[1] ✉, **Paul Zaslansky** ®[2], **Chloë Boote**[3], **Annabelle Doheny**[3], **Tali Mass** ®[1,4] **& Emma F. Camp** ®[3,4] ✉

The alarming rate of climate change demands new management strategies to protect coral reefs. Environments such as mangrove lagoons, characterized by extreme variations in multiple abiotic factors, are viewed as potential sources of stress-tolerant corals for strategies such as assisted evolution and coral propagation. However, biological trade-offs for adaptation to such extremes are poorly known. Here, we investigate the reef-building coral *Porites lutea* thriving in both mangrove and reef sites and show that stress-tolerance comes with compromises in genetic and energetic mechanisms and skeletal characteristics. We observe reduced genetic diversity and gene expression variability in mangrove corals, a disadvantage under future harsher selective pressure. We find reduced density, thickness and higher porosity in coral skeletons from mangroves, symptoms of metabolic energy redirection to stress response functions. These findings demonstrate the need for caution when utilizing stress-tolerant corals in human interventions, as current survival in extremes may compromise future competitive fitness.

Climate change is threatening the future of coral reefs globally. Ocean warming, seawater deoxygenation, and ocean acidification are the primary stressors that coral reefs face as climate changes. Between 2009 and 2018 alone, 14% of global coral cover was lost[1]. These declines are predicted to continue in the short-term even under the most optimistic scenarios of emissions reduction[2]. Altered reef conditions presage that traditional reef management is not sufficient to protect coral reefs from the mounting stressors that they are facing. While reducing carbon emissions is fundamentally necessary[3], researchers are exploring innovative active management opportunities that may assist future coral survival[4]. One approach has been to identify stress-tolerant coral populations that survive in naturally extreme coral environments[5]. These corals have the potential to be used in active management, e.g. through direct propagation and out planting, while also providing new knowledge about the mechanisms that support stress tolerance in corals[6].

Corals that grow in mangrove lagoons where the seawater is warmer, more acidic, and has lower oxygen conditions than

neighboring reef sites, have become important natural laboratories to advance our understanding of coral resilience and the associated fitness trade-offs at multi-stressor locations[7]. To date, extreme mangrove systems have been identified at sites in the Caribbean, the Seychelles, Indonesia, New Caledonia, and the Great Barrier Reef[8–11]. These areas are highly heterogeneous, but common described mechanisms that support coral survival include unique associated microorganisms (both bacteria and Symbiodiniaceae[8,12–14]), altered physiology[8,15], and divergent photosynthetic strategies[8,16]. Mangrove locations have been proposed to precondition resident corals to climate change[10] and laboratory studies have demonstrated that corals from mangrove lagoons have higher calcification rates under low pH[13] as compared with their reef counterparts. However, under ambient conditions, a trade-off of reduced calcification rates has been documented in these mangrove lagoons[8,9]. Whether reduced mineralization rates are accompanied by other adjustments in skeletal structural properties, such as density and hardness, remains unknown. Furthermore, what molecular changes accompanying coral survival in extreme mangrove

[1]Department of Marine Biology, Leon H, Charney school of Marine Sciences, University of Haifa, Haifa, Israel. [2]Department for Operative, Preventive and Pediatric Dentistry, Charité-Universitätsmedizin, Berlin, Germany. [3]Climate Change Cluster, University of Technology Sydney, Ultimo, NSW, Australia. [4]These authors contributed equally: Tali Mass, Emma F. Camp. ✉e-mail: fscucchia@ufl.edu; emma.camp@uts.edu.au

locations have yet to be described, but it has been hypothesized that reduced genetic diversity due to environmental selection or restricted gene flow may occur[7].

To date, in the mangrove systems studied, reduced species diversity relative to neighboring reefs is a common finding[9,10], though not always observed[17]. The genus *Porites* is considered a hardy coral taxon and has been documented across extreme mangrove systems globally[10,17,18]. While some *Porites* species are weedy, for example *P. astreoides* in the Caribbean[19], other species such as *P. lutea* in the Indo-Pacific are important reef-forming species[20]. *P. lutea* is a dominant coral at the Woody Isles mangrove lagoon on the Great Barrier Reef[8]. Corals at Woody Isles experience dynamic diel and seasonal changes in temperature (range >7.7 °C), pH (1.3), oxygen (7.33 mg/L), and salinity (15.5 ppt) compared to the more stable physicochemical conditions of the adjacent (<500 m away) Low Isles reef[8,21]. Such changes in the abiotic landscape influence symbiotic partnerships of the corals, as different species of the C15 Symbiodiniaceae have been detected in *P. lutea* native to Woody Isles (single genotype; ITS2 type profile C15-C15by-C15bn) compared to colonies on Low Isles reef (three identified genotypes; C15, C15-C15bp-C15l-C15n-C15.8, and C15-C15l-C15n)[8]. These unique symbiont associations of *P. lutea* at Woody Isles correspond with different photosynthetic energy dissipation strategies (i.e., differential investment in photochemical versus non-photochemical quenching), reduced net photosynthesis, and enhanced respiration[8]. Both light and dark calcification have also been reported to be reduced for *P. lutea* at Woody Isles (by 30% and 22% respectively[8]).

Given the hypothesized value of extreme locations such as the Woody Isles, as potential refugia and sources of stress-hardened corals[6], we sought to understand whether there are underlying genetic and morphological trade-offs that could compromise the suitability of these sites for coral conservation. Here, we apply transcriptome-wide single nucleotide polymorphism (SNPs) detection to test whether *P. lutea* colonies from Woody Isles are genetically distinct from the ones on the neighboring reef found <500 m away (Low Isles), and if they possess lower genetic variability than their reef counterparts. We couple this tier of exploration with the gene expression profiling of *P. lutea* at the two locations to investigate key biological processes associated with living in extreme mangrove environments. Further, we employ a combination of synchrotron X-Ray tomography, electron microscopy, skeletal morphometrics, and material structural analyses to assess whether skeletal properties that determine structural integrity are similar across environments, in light of previous reports of reductions in calcification rates in mangrove locations[8,9]. With our combination of molecular and morphological techniques, we show trade-offs between genetics, energetics, and skeleton development in corals living in a naturally extreme environment. These findings shed light on the costs of environmental stress hardening for coral reef management in the face of climate change.

## Results

### Genetic differentiation and gene expression dynamics across habitats

Clustering based on the proportion of shared alleles (Fig. 1a) and genetic distance measurements ($F$st = 0.051, $p < 0.05$) show significant genetic differentiation between reef and mangroves *P. lutea* corals. Intra-habitat variability indicates that reef corals are more genetically heterogeneous than mangroves corals (Fig. 1a), a pattern that is reflected in the clustering based on gene expression dynamics (Fig. 1b). Whereas habitat origin explains most of the variation in gene expression (PC1, 72%), reef corals show a higher inter-individual variability in gene expression compared to mangroves corals (Fig. 1b).

All differentially expressed genes (DEGs) with similar expression profiles (2713 up-regulated, 2560 downregulated; Supplemental Fig. 1) grouped into two k-means clusters (Fig. 2a): genes in cluster 1 are up-regulated in mangroves corals and down-regulated in reef corals,

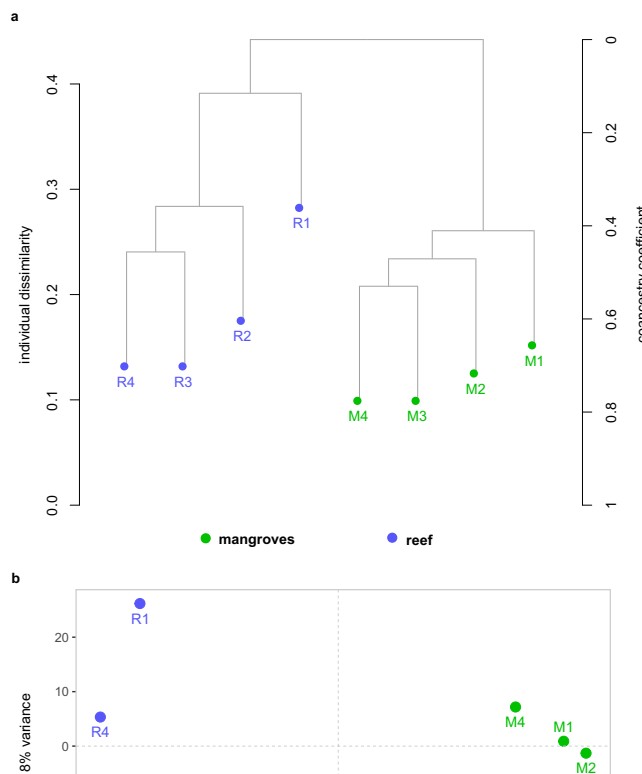

**Fig. 1 | Genetic differentiation and gene expression patterns between habitats. a** Hierarchical clustering dendrogram based on proportions of shared alleles shows genetic differentiation between reef and mangrove corals. Branch height represents dissimilarity scores on the left and the coancestry coefficient on the right, based on pairwise identity-by-state distances among samples. **b** Habitat of origin explains most of the variations between the gene expression profiles of reef and mangrove corals (PC1). The Principal Component Analysis is based on variation in mean log2 fold-change across differentially expressed genes.

whereas genes in cluster 2 exhibit the opposite pattern. To provide an overview of the biological processes with differential activity between reef and mangrove corals, we performed a Gene Ontology (GO) enrichment analysis and found 940 GO terms in cluster 1 and 156 in cluster 2 (Supplemental Data 1). These are summarized by 38 GO slims (Fig. 2b), which include functions related to cellular stress response (cytoskeleton organization[22,23], ribosome biogenesis[24]), regulation of cellular respiration (mitochondrial gene expression[25]) and oxidative stress response (metal ion homeostasis[26], oxidoreductase activity functions within the catalytic activity GO slim, Supplemental Data 1) that have higher activity in mangroves corals compared to reef corals (enriched in cluster 1).

### Morphological adjustments to an extreme environment

Considering that components of stress responses need to be fueled by energy at the molecular, cellular, and systemic levels[27], we examined potential trade-offs in coral growth associated with living in energy-demanding highly variable environments, such as mangrove forests. Skeletal characteristics of mangrove corals include significantly higher bulk porosity (Mann-Whitney test, $p < 0.01$) and lower density (Mann–Whitney test, $p < 0.01$) compared to reef corals (Fig. 3a, b, e), also reflected by the significantly lower thickness of the skeletal walls

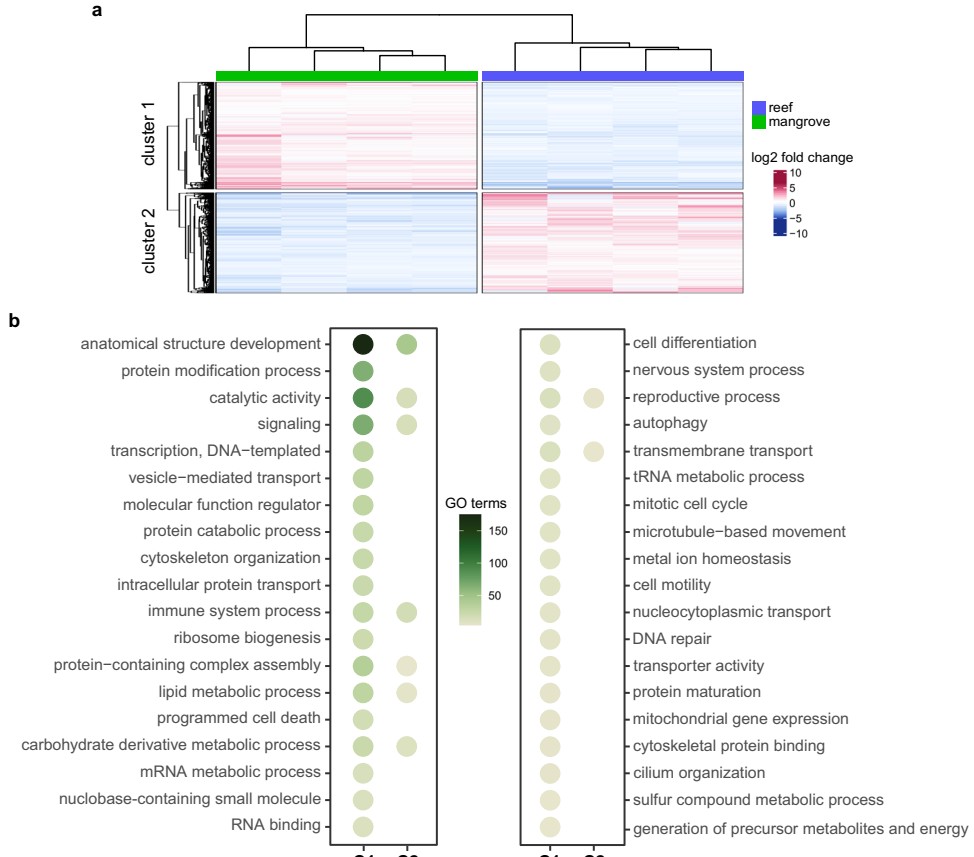

**Fig. 2 | Gene expression profiles and enriched biological processes between reef and mangrove corals. a** Differentially expressed genes with similar expression profiles grouped into two distinct k-means clusters. Clustering is based on expression ($\log_2$ fold-change) relative to the row mean in the gene count matrix. **b** Biological functions enriched within each gene expression cluster. Each Gene Ontology (GO) slim category summarizes a group of significantly (Wallenius test, $p < 0.05$) enriched GO terms (list available in Supplemental Data 1). Dot colors represent the number of GO terms enriched within each GO slim. For graphical purposes, only GO slims with more than three enriched GO terms were plotted. C1 cluster 1, C2 cluster 2.

of the single coral polyps within the colony of mangrove corals (unpaired $t$ test, $F = 1.542$, $p < 0.05$; Fig. 3c–e). Such differences in skeletal morphometrics do not appear to affect the material structural properties, as we found no difference in micro-indentation hardness between the two locations (Fig. 3e and Supplemental Fig. 2). Electron microscopy revealed that mangrove corals possess smaller calyxes (unpaired $t$ test, $F = 2.287$, $p < 0.0001$; Fig. 4a–c) and a deeper distribution of the spines which differs from the shallower spine arrangement of reef corals (Fig. 4d, e). In addition, we observed a less compact arrangement of the skeletal fibers of mangrove corals, as compared with the smoother surface observed in reef corals (Fig. 4f–i).

The observed morphological differences suggest underlying genomic regulation of the biomineralization process. We indeed found differential expression of several biomineralization-related genes between the two habitats, with mangrove corals mostly exhibiting down-regulation of genes coding for skeletal organic matrix proteins, as compared to reef corals (Supplemental Fig. 3).

## Discussion

Coral reefs continue to decline at an alarming rate, a trajectory that is anticipated to continue in the face of intensifying climate change[2]. For sessile invertebrates such as corals that have limited ability to move to more suitable conditions, researchers have been exploring the possibility of utilizing more resilient 'super corals' in active reef management intervention[5,6] to secure a future for coral reefs. Mangrove systems have been highlighted as a potential source for such resilient corals, because of their survival in waters where multiple abiotic

factors covary creating environmental conditions similar to what future reefs are predicted to experience. There is however an urgent need to evaluate what biological trade-offs come with survival and stress-hardening in such naturally extreme environments, to allow informed discussion about the ecological risks associated with harnessing resilient varieties of key habitat-forming organisms for active intervention strategies. Our work shows that coral stress-hardening comes with compromises in genetic, energetic, and morphological characteristics.

We revealed genetic differentiation among conspecific corals (Fig. 1a) from neighboring mangrove and coral reefs, likely attributable to the extreme nature of mangrove habitats where environmental variability constrains gene flow between populations, as previously shown[28]. Harsh environments also elicit strong selective pressures that decrease intra-population genetic diversity[29]. Indeed, we find lower genetic heterogeneity among mangrove corals (Fig. 1a), which mirrors the lower inter-individual variability in gene expression compared to reef corals (Fig. 1b). Genetic variation between individuals can influence the magnitude of gene expression[30,31]. Both the amount of genetic and gene expression variations are important factors for the ability of organisms to rapidly adapt[32,33]. In fact, variability in gene expression levels represents an advantage that favors adaptation to environmental changes and or/stress conditions[34], and thus, mangrove corals may be at a disadvantage in the future as conditions become even more extreme and unpredictable due to global climate change. While mangrove corals have adapted to the present-day extreme environmental conditions, if other stressors that are predicted to intensify,

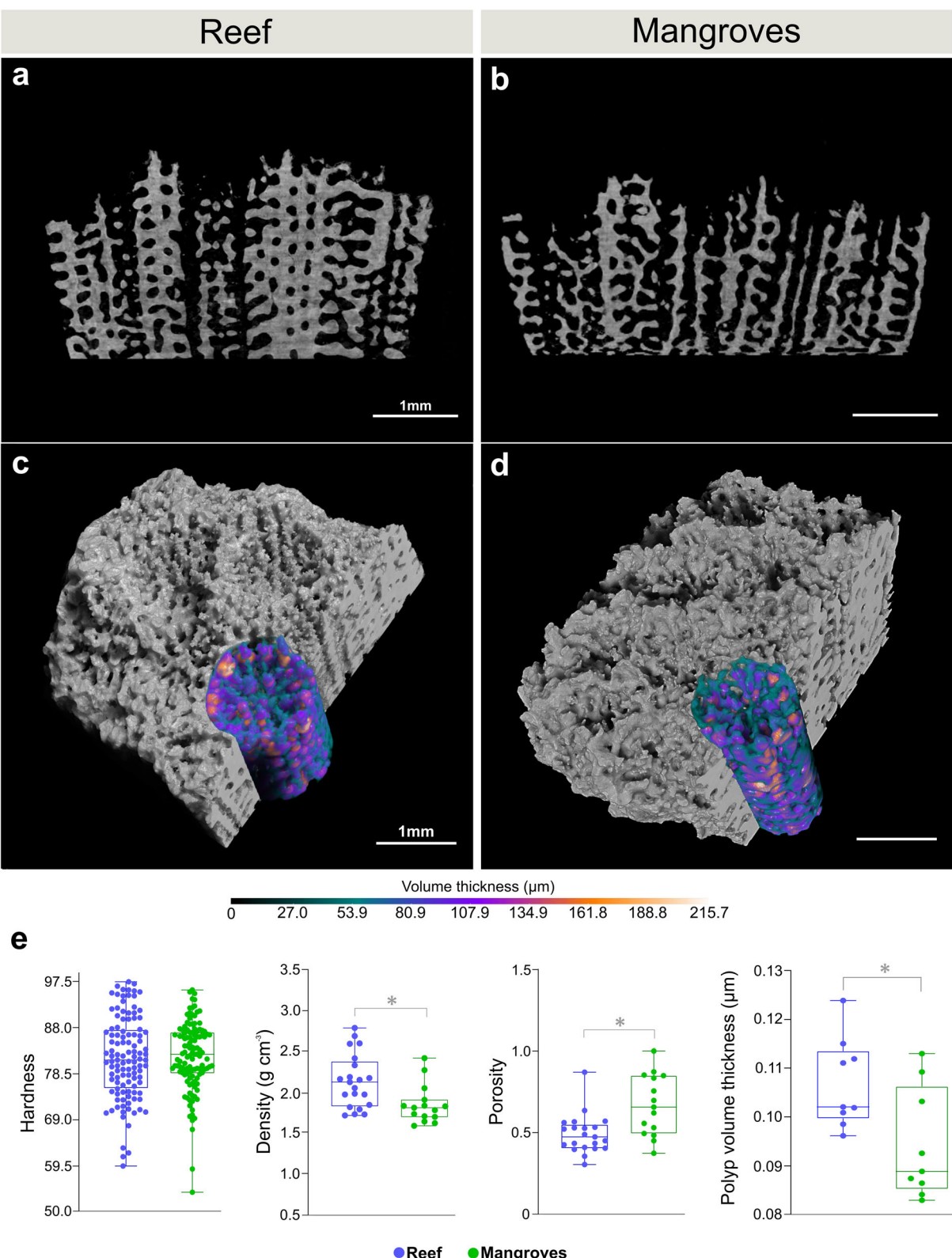

**Fig. 3 | Changes in skeletal morphometrics among locations. a**, **b** Cross-sectional tomography slices providing an overview of the internal skeleton architecture, which was characterized in terms of (**c**, **d**) single-polyp volumetric thickness, (**e**) hardness (shore hardness value, higher values indicate a harder material), density, and porosity (ratio of the pore volume connected to the external skeleton surface to bulk volume of the skeleton). Boxplots in **e** show first and third quartiles, median line, and whiskers at ±1.5 interquartile range. Asterisks represent significant differences computed with two-sided unpaired $t$ test (polyp volume thickness, $n = 9$, $F = 1.542$, $p < 0.05$) or two-sided Mann–Whitney test (density, $n = 15$–$21$, $p < 0.01$; porosity, $n = 15$–$21$, $p < 0.01$). No significant difference in hardness was found between mangroves and reef corals (two-sided Mann–Whitney test, $n = 120$, $p > 0.05$). Source data are provided as a Source Data file.

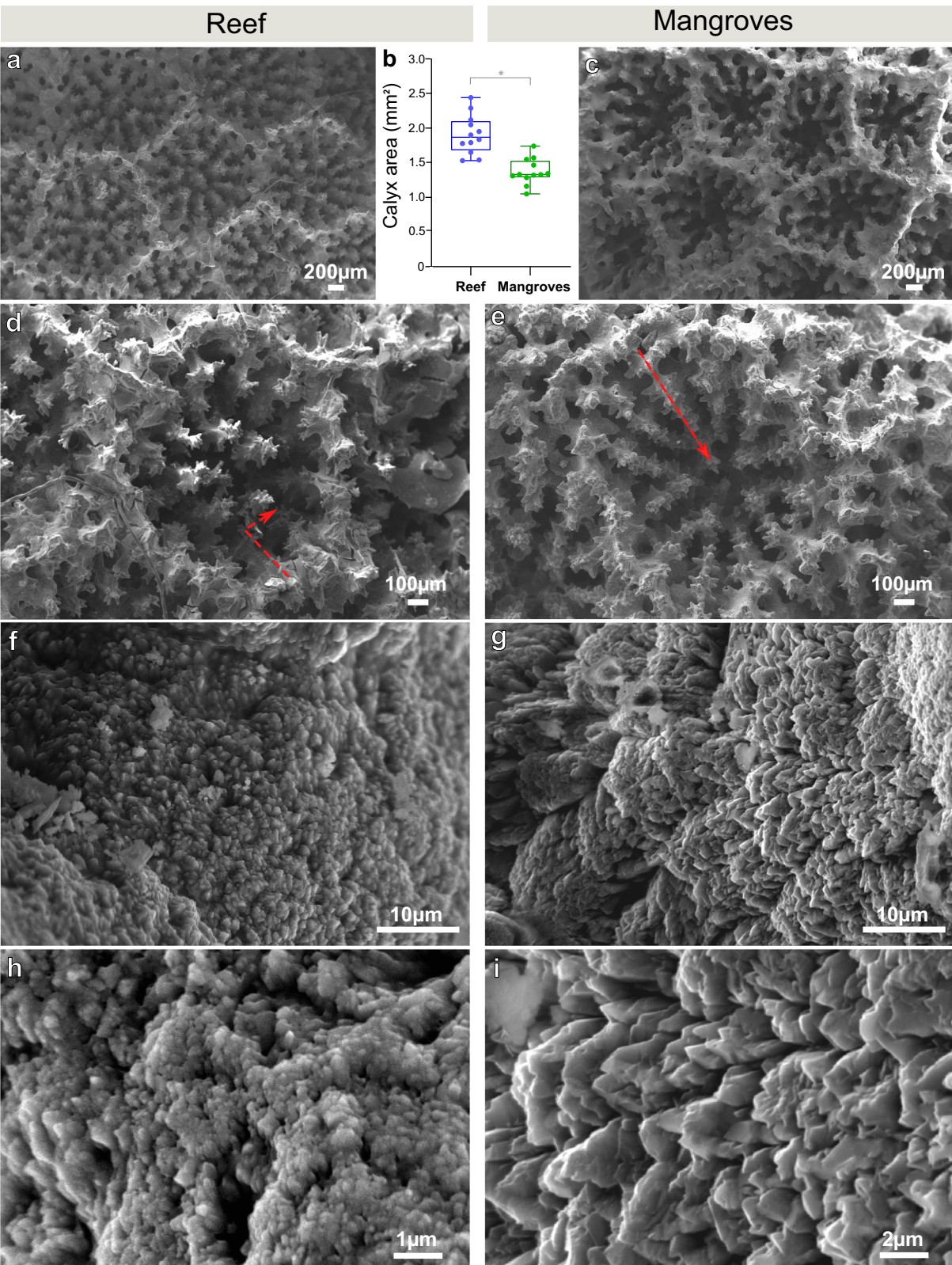

**Fig. 4 | Differences in micro-skeletal features of *Porites lutea*. a, c** Surface morphology by electron microscopy. **b** Polyps of reef corals have larger calyxes compared to mangrove corals. Boxplots in **b** show first and third quartiles, median line and whiskers at ±1.5 interquartile range, asterisk represents a significant difference computed with two-sided unpaired *t* test ($n = 12$, $F = 2.287$, $p < 0.0001$). **d, e** Within each single calyx, spines are arranged in a downward configuration in mangrove corals which differs from the shallower arrangement of reef corals (indicated by the red arrows). **f, g** Enlargements of the spines showing granulated surface composed by **h, i** bundles of fibers with a less compact and less rounded texture in mangrove corals compared to reef corals. Spines arrangement and surface texture observations were repeated independently for $n = 12$ polyps per each habitat. Source data are provided as a Source Data file.

such as disease[35], impact the mangrove corals, they could be more at risk due to the smaller pool of alleles and gene expression variations compared to the more genetically heterogeneous reef corals. It must also be noted that the higher coral species richness observed in some mangrove locations compared to the reef sites[17] may mask the underlying intra-species low genetic richness.

Our small sample size may have only limited power for adequate diagnosis of the genetic variation between populations. Yet, there is wide variability in sample size used for population genomic work, indicating that there is no universal sample size rule allowing to accurately depict the majority of genetic variation across populations and taxa (reviewed in ref. 36). However, it recurrently appears that even with restricted sample sizes (<10) it is possible to adequately differentiate between populations[36–41], especially in the case of populations with confined geographical distributions[36], such as for corals living in mangrove lagoons and adjacent reefs. Multiple studies have also demonstrated that increasing the sample size does not improve, or only minorly improves, accuracy of the genetic diversity estimation[37,40,42,43]. Moreover, in the case of very small sample size, screening large numbers of SNPs (>1500) can produce accurate estimates of genetic diversity[39,40,44,45].

The fluctuating and extreme environmental conditions of mangroves lagoons drive the up-regulation of genes linked to cellular stress responses and to the regulation of cellular respiration (Fig. 2b), recapitulating previously observed patterns of enhanced respiration rates in mangrove environments compared to the reef[8,9]. Biological processes including basic maintenance of vital functions and response to daily stressors, require a considerable amount of energy, largely met by mitochondrial respiration[27]. A substantial flux of energy towards maintenance of physiological integrity under stressful conditions, combined with reductions in net photosynthesis[8,9], come at the cost of reduced calcification rates in mangrove corals[8,9]. Here we show that such reduction is underlined by previously unseen variations in skeletal morphometrics, such as lower bulk skeletal density and higher porosity compared to reef corals (Fig. 3e). This pattern, mirrored by the reduced thickness of the skeletal walls at the single-polyp level (Fig. 3c–e), can be considered a gain for corals living in mangrove habitats. In fact, similar to naturally acidic $CO_2$ vent environments, higher bulk porosity facilitates maintaining linear extension rates constant to potentially meet functional reproductive needs (i.e., to reach the critical size required for sexual maturity)[46]. In addition, compared to reef sites, mangrove lagoons are not subjected to strong wave action and currents, thus the skeleton morphotype of mangrove corals may be the product of local adaptation to a more protected location. In fact, water motion can influence the skeleton development dynamics of stony corals, with lower flow intensity leading to the formation of less compact and reduced thickness skeletons[47,48]. However, if on one hand a less dense and more porous skeleton is favored within a mangrove lagoon system, such a trait may represent a disadvantage in the case of coral transplantation to reef sites, since higher porosity and lower density could increase damage susceptibility due to wave action. Thus, transplanting stress-hardened corals originating from multi-stressor extreme systems, such as mangrove lagoons, to help future-proof reef sites may not be as beneficial as hoped. In other variable extreme sites, like the semi-enclosed bays of the Palauan Rocky Islands, not all coral species show trade-offs between environmental tolerance and skeleton growth[49], suggesting that compromises between different biological traits to support stress resistance are not ubiquitous among corals. There is however a more complex biological landscape underlying the variety of strategies adopted by corals to cope with multi-stressor environments, including changes and trade-offs in physiological processes that may have been overlooked[49].

Increased skeleton porosity and lower density compromise mechanical resistance. Yet, we found no difference in skeletal hardness between mangrove and reef corals (Fig. 3e), suggesting that other morphological properties determine the skeleton mechanical behavior.

In fact, this is strongly controlled by the architectural arrangement of the basic building blocks of the biomineralization process, that is the aragonite bundles of fibers[50], which possess a different architecture between mangroves and reef corals (Fig. 4f–i). The spacing of voids that results from the arrangement of the structural units of the skeleton can also confer added strength to the coral colony[51]. Here we find clear differences between mangrove and reef corals in the spatial organization of several skeletal components, including the size of the calyx and the arrangement of the septal spines (Fig. 4a–e). Differences in such morphological features in corals are implicated in the regulation of within-colony light levels to optimize endosymbiont light harvesting and photosynthesis[52–54]. Moreover, the microgeometry of the coral skeletal surface appears to regulate networks of currents at the polyps surface, controlling sharing of nutrient, $O_2$ and metabolites generated by the algal endosymbionts[55]. Changes in surface skeletal features, such as those observed here (Fig. 4a–e), may therefore contribute to complement the endosymbiont photosynthetic activity under the flow conditions of mangrove and reef locations. Corals in these sites, in fact, host different species of algal symbionts[8] that have specialized to live either in mangrove or reef environments. This implies that it is not only the photosynthetic strategy and flexibility of the algal endosymbionts that supports the coral capability to thrive from reef to mangrove sites[8], but also host skeletal adjustments, such as those observed here, play a substantial role.

Taking a broader look at our findings and to the results of previous works (reviewed in ref. 7), it becomes evident that corals living in extreme environments possess common traits representing adaptations and/or acclimation mechanisms to changes in abiotic features. The extreme nature of such habitats incurs trade-offs among the various functions that contribute to survival, because of the competing demands on metabolism. For corals, enhanced tolerance to a particular abiotic stressor generally leads to reduced growth, lower skeletal density and/or reproductive output (see ref. 7 and references therein). In mangrove lagoons, the unique environmental settings have produced clear trade-offs between increased tolerance to a combination of multiple co-varying stressors and growth (Fig. 5), as well as driving a strong selective pressure reducing genetic differentiation of local coral species (Fig. 5). As such, the present superiority in coping with fluctuating stress events of mangrove corals may not guarantee the long term resilience they have been claimed to possess[6,11] under future harsh environmental conditions. Caution must therefore be undertaken when considering their role in active management. Nonetheless, because of the resemblance to climate change scenarios, mangrove lagoons still yield important lessons for the management of coral reefs. In fact, other than informing us on what taxa are more likely to survive into the future and what biological traits they will be characterized of, they inform us of the critical need to preserve current global coral genetic richness, to ensure not only coral survival but also the maintenance of key ecosystem attributes and processes.

## Methods
### Site description and sample collection
This research complies with all relevant ethical and permit regulations of the University of Technology Sydney, the University of Haifa, and the Great Barrier Marine Park Authority. The Woody Isles site (Fig. 6) is a semi-enclosed lagoon surrounded by mangrove forest that undergoes daily tidal flushing. The shallow (0–2.5 m) nature of the lagoon in combination with rich nutrient cycling and associated microbial activity results in highly variable abiotic conditions that are characteristic of mangrove systems (Supplemental Table 1). The Low Isles reef site (Fig. 6) experiences relatively stable abiotic conditions[8] and has ca. 35% coral cover that is dominated by *Porites* sp. (AIMS Long Term monitoring dataset [https://apps.aims.gov.au/reef-monitoring/reef/16028S/benthic]). Similar mean daily light levels have been previously documented at both Woody Isles and Low Isles during the winter[8].

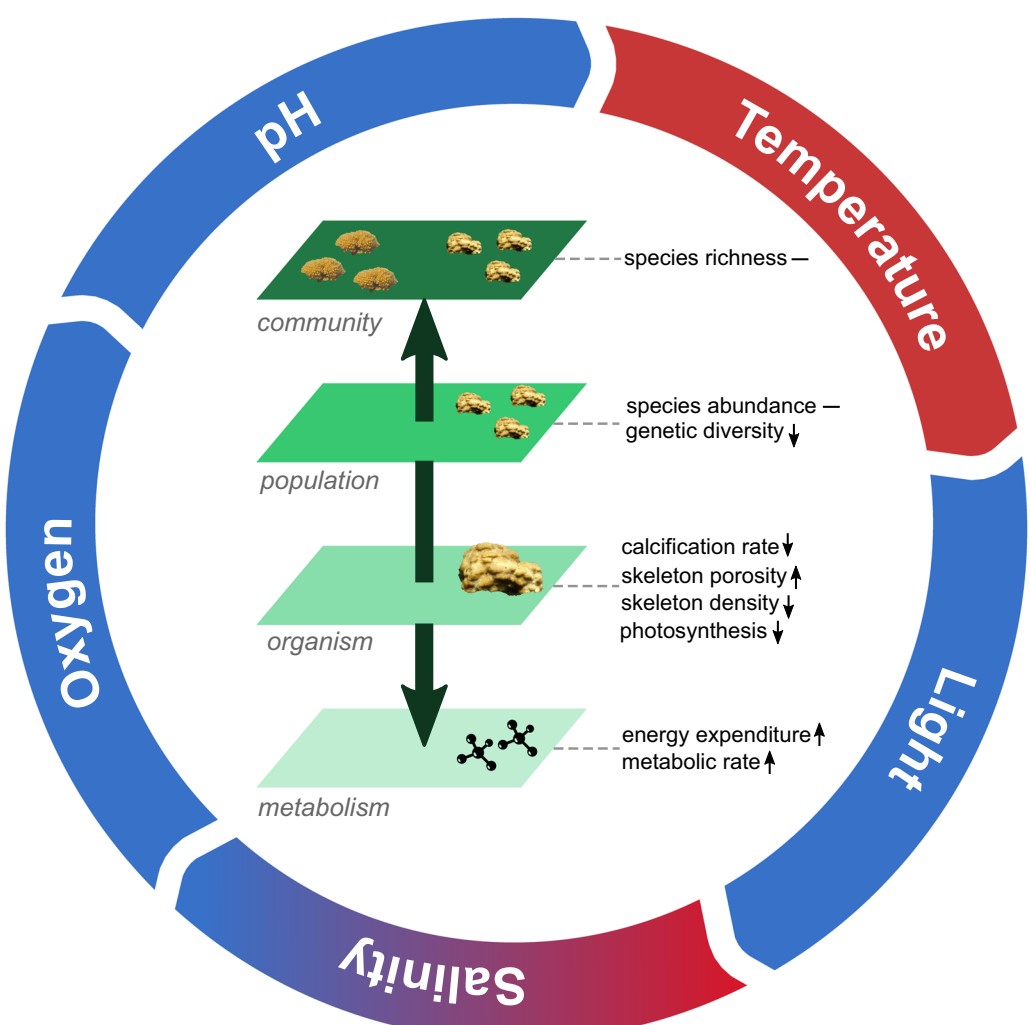

**Fig. 5 | Conceptual diagram of coral trade-offs across organismal to ecological scales associated with living in an extreme multi-stressor environment.** Common dynamics in biological traits observed in corals living in mangrove environments, characterized of higher temperatures (red color in the scheme), higher or lower salinity levels (red and blue color together), and lower oxygen, light, and pH levels (blue color) compared to neighboring reef sites[8–11,13–15,17,18,97,98] (see Supplemental Table 1). To cope with such climate change like fluctuations in abiotic environmental features, corals put in place trade-offs spanning between different scales of biological organization (organismal to population levels). For example, higher metabolic rates, e.g. coral host respiration rates, and energy expenditure (upward arrows) leads to reduction in calcification rates and skeleton density (downward arrows). Such environmental setting also drives strong selective pressure which tends to reduce intra-population genetic diversity. At higher levels of biological hierarchy, both increases and decreases in species abundance and richness have been observed between different coral species (indicated by a slash).

Corals at the reef and mangrove sites were sampled at a depth of 1–1.5 m. Colonies (9-10 per site) of comparable size were sampled with a chisel (5–6 m apart to minimize the potential of sampling clonal genotypes[56]) to obtain small colonies (4–6 cm total length) at both sites. Samples were placed in a zip-lock bag containing native seawater to return to the research vessel (<20 min). Once on the research vessel, colonies were assigned to either molecular analysis (4 per site) or skeleton physical property analysis (5–6 per site). Fragments for molecular analysis were immediately frozen in liquid nitrogen and maintained at −80 °C at the University of Technology Sydney prior to RNA extraction. Fragments for physical property analysis were air-dried and stored in protective containers to prevent corallite damage prior to physical property analysis. All sampling was conducted under the Great Barrier Marine Park Authority permit G18/40023.1.

### RNA extraction, sequencing, and read processing

Snap frozen *P. lutea* were fragmented into small pieces of tissue and placed in 2 mL microtubes containing 500 µl of DNA/RNA shield (Zymo Research, Catalog No R1100-250). Total RNA was extracted using the RNeasy mini kit according to the manufacturer's protocol (Qiagen, Catalog No 74104). RNA presence was confirmed using gel electrophoresis. RNA concentration and quality were tested on a NanoDrop (Thermo Fisher Scientific) and TapeStation (Agilent Technologies), respectively. Four independent samples were obtained for each habitat. RNA-seq libraries were prepared at the AGRF sequencing facility (Australia) and sequenced using an Illumina Stranded mRNA workflow involving polyA selection from 500 ng of total RNA. Approximately 749 million total 150 bp paired-ends reads (47 ± 4 million reads per sample, reported as mean ± SEM) were sequenced on an Illumina NovaSeq 6000. Raw reads were trimmed and filtered using Cutadapt (v2.6[57]), Trimmomatic (v0.39[58]), and SortMeRNA (v4.3.6[59]), to remove adapters, low-quality reads, and ribosomal RNA. Prior alignment to the reference genome, we ensured that all samples belonged to *P. lutea* by subjecting cleaned reads to a BLASTx search using Diamond (v2.0.11[60]) against open-access proteome databases of stony corals (Supplemental Table 2). Across all samples, 75% ± 0.04% (mean ± s.e.m.) of all reads belonged to *P. lutea*. Additionally, we identified the dominant

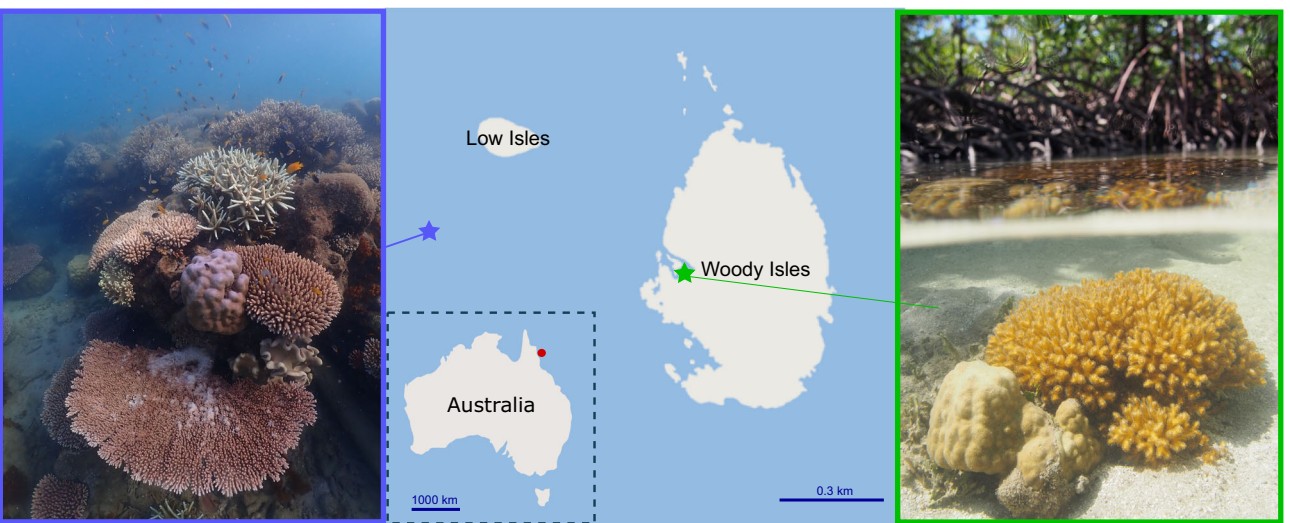

**Fig. 6 | Study sites.** Overview of the reef (left image) and mangrove (right image) sites on the Great Barrier Reef adjacent to Low Isles and Woody Isles, respectively, where *P. lutea* corals were collected. Pictures provide a typical view of *P. lutea* found in these two environments. The red dot off the coast of Australia indicates the location of the study area.

algal symbionts BLASTing the cleaned reads against open-access proteome databases of Symbiodiniaceae species (Supplemental Table 2), and found 69% ± 0.06% (mean ± s.e.m.) of all reads belonging to *Cladocopium* spp., in line with previous observations in mangroves and reef corals in the same area[8].

Reads were then aligned to the *P. lutea* host genome assembly[61] using HISAT2 (v2.2.1[62]) in the stranded paired-end mode and assembled using StringTie (v2.2.5[63]). GFFcompare (v0.12.2[64]) was used to assess the precision of mapping by comparison of merged mapped GFFs to the *P. lutea* reference assembly. Finally, a gene count matrix was generated using the StringTie python script prepDE[63]. To provide gene ontology (GO) annotation for functional enrichment analysis, functional annotation of *P. lutea* protein sequences[61] was performed using Diamond (v2.0.11[60]), Uniprot[65], and eggNOG[66].

### Differential expression and functional enrichment analyses

The gene count matrix from StringTie was imported within the R environment (v3.6.3[67]) and filtered to remove low-counts genes (genes with less than 10 counts in at least 4 of 8 samples were excluded). Differential expression analysis with DESeq2 (v1.26.0[68]) was performed between habitats using the Wald test[69], to identify differentially expressed genes (DEGs) with adjusted p-value (FDR, <0.05). Counts were then normalized for visualization using the variance stabilizing transformation in DESeq2 (v1.26.0[68]) and gene expression patterns were examined with PCA, by calculating sample-to-sample distances, and with heatmap and k-means clustering using the R package NbClust (v3.0[70]). Expression of known coral biomineralization-related proteins (reported in ref. 71 and available in Supplemental Data 2) was tested against the differential expression results, performing a BLAST search to identify the biomineralization genes in the genome of *P. lutea*[61].

GO enrichment analysis was performed per each NbClust k-mean cluster using the R package Goseq (v1.42.0[72]). All genes analyzed in DESeq2 were used as foreground group and all significant differentially expressed genes as background. Enrichment analysis was performed using the default 'Wallenius' model, and statistically significant enriched GO terms were selected with a cutoff of *p* < 0.05. For the resulting enriched GO terms, slim categories were obtained with the function 'goSlim' of the R package GSEABase (v1.52.1) using the GOslim generic obo as reference database (v1.2[73]). We used GO slims to give a broad overview of the GO functional distribution, summarizing the processes or functions that are enriched in mangroves and reef corals. The full list of enriched GO terms per each slim category is available in Supplemental Data 1.

### Genotyping and assessment of genetic differentiation

Single Nucleotide Polymorphisms (SNPs) were extracted from the cleaned reads using the Genome Analysis Toolkit framework (GATK, v4.2.0; ref. 74) following the recommended RNA-Seq SNPs practice of the Broad Institute[75]. First, HISAT-aligned reads were sorted and marked for duplicates, then variant calling was performed with the GATK HaplotypeCaller tool[74] and genotypes were finally jointly called using GATK GenotypeGVCFs. The joined variant-calling matrices were filtered for quality by depth and for linkage disequilibrium using the --indep-pairwise function of PLINK (v2.0[76]), identifying 22,925 SNPs, which fall in the range of other population genomic studies on corals[77–81] and other organisms[37,40,41,43–45,82–85] that employ <30,000 SNPs to assess genetic differentiation.

To determine genetic differentiation between individuals and habitats, we inferred the individual dissimilarity coefficient and the proportions of shared ancestral alleles by computing pairwise identity-by-state distances using the SNPRelate R package (v1.20.1[86]). A final *n* x *n* matrix of 8 individuals and a total of 80,725 genotypes (0,1,2) was employed for hierarchical clustering, conducted with the snpgdsHCluster function using 999 permutations. Additionally, we assessed the genetic differentiation between reef and mangroves corals by estimating the fixation index ($F_{ST}$[87]) using the package HIERFSTAT (v0.5.10). Resulting $F_{ST}$ pairwise values were tested for significance ($p < 0.05$) with 999 permutations.

### Skeletal macro- and micro-morphology

Multiple tests were performed to assess the physical properties of the coral skeletons. Some of the tests were destructive, while others were not, and thus sample numbers varied between tests. All skeletal parameters were consistently measured from the top portion of the colony across all replicates and for both habitats. Bulk density and porosity were determined for 5–6 samples per habitat (three technical replicates per sample) following the methods described in ref. 46. Specifically, the buoyant method, which is based on the Archimedean principle, was used to measure the total volume occupied by the coral skeleton (bulk volume, $V_B$), the bulk density (ratio of the mass to bulk

volume, $d_b$) and the effective porosity ($P_A$, ratio of the pore volume connected to the external surface ($V_A$) to bulk volume, $P_A$). First, a precision balance was used to weigh the skeletons to determine the dry mass ($m$), and then skeletons were placed inside a dry chamber and evacuated. This was followed by gentle introduction of water to fully saturate the samples. The chamber was then vented to reach ambient atmosphere, and the mass of the fully water-saturated skeletons ($m_s$) was determined. The mass of saturated skeletons fully immersed in water ($m_h$) was measured with a hydrostatic balance. Skeletal parameters were calculated using the following equations: $V_A = (m_s - m)/\rho_w$, $V_B = (m_s - m_h)/\rho_w$, $P_A = V_A/V_B$, $d_b = m/V_B$, in which $\rho_w$ = water density.

For hardness, a ~1 cm cross section of coral was taken from the center of each sample (same location across samples, $n = 3$ samples per study site). Cross sections were embedded in resin and polished[88], before micro-hardness (40 replicate measurements per sample; 20 from the external 5 mm and 20 from the internal 5 mm to account for differences in old versus new growth) was assessed using a Model D Shore scleroscope, calibrated on reference material and optimized for natural rock as outlined by International Society for Rock Mechanics (ISRM, 2014). This method assesses hardness as the rebound height of the diamond-tipped hammer, by its own weight and from a fixed height. It is measured on a calibrated scale which gives the Shore hardness value (SH) in its own units, with higher values indicating a harder material[89]. A section adjacent to the hardness sample cross section was used for SEM, where fragments were vacuum coated with gold (for conductivity) prior to examination under a ZEISS SigmaTM SEM (Germany) SE2 detector (5–15 kV, WD = 4–15 mm). In each fragment, the area of the calyx of individual polyps ($n = 12$) was measured with FIJI[90].

Skeletal parameters were tested for normality (Shapiro-Wilk test) and homogeneity of variance (Levene's test). Statistical significance ($p < 0.05$) of differences between habitats was assessed with parametric unpaired t-test (calyx area, hardness between old and new growth) and non-parametric Mann–Whitney test (hardness–old and new growth analyzed together, porosity and density) using the GraphPad Prism software (v9.0.0)(GraphPad Inc., San Diego, CA, USA).

## Polyp skeletal thickness

Tomographic scanning of the skeleton fragments was conducted at BAMLine[91], the imaging beamline of the synchrotron electron storage ring BESSY II operated by the Helmholtz-Zentrum Berlin für Materialien und Energie (HZB), Berlin, Germany. Each sample ($n = 3$ per habitat) was scanned with incremental rotation (multiple projections spanning 360°) on a high-resolution imaging sample stage[92] with exposure times set to 140 ms. Projection images were acquired using an energy of 24.5 keV with an effective pixel size of 3.61 μm.

Data normalization and reconstruction were performed according to the steps detailed in ref. 93. In brief, data were normalized to account for beam inhomogeneities by background-correcting the radiograms with best-fitting (minimum variance) empty beam (flat-field) images, and by subtraction of dark-current images. Reconstruction was performed by the filtered back projection method using nRecon (v 1.7.4.2, Brucker micro-CT, Kontich, Belgium) to generate 3D datasets from cross-sectional 2D images. Tomographic datasets were visualized and processed in 3D using Dragonfly (v2021.3, Object Research Systems-ORS, Montreal, Quebec, Canada)[94]. Volumetric thickness of the skeleton walls of digitally isolated single coral polyps ($n = 9$ per habitat) was computed using the "Create Volume Thickness Map" function built into Dragonfly, which performs volume thickness measurements based on the sphere-fitting method (Supplemental Fig. 4 and Supplemental Movie 1). Statistical significance ($p < 0.05$) of differences in mean volume thickness values between habitats was assessed using unpaired t-test, after testing for normality (Shapiro-Wilk test) and homogeneity of variance (Levene's test), with the GraphPad Prism software (v9.0.0)(GraphPad Inc., San Diego, CA, USA).

## Reporting summary

Further information on research design is available in the Nature Portfolio Reporting Summary linked to this article.

## Data availability

The morphological data generated in this study have been deposited in the Zenodo database[95] (https://doi.org/10.5281/zenodo.7454382). Transcriptomic data generated in this study have been deposited in the National Center for Biotechnology Information under accession code PRJNA912580. Morphological and genetic data generated are also provided in the Supplementary Information. Source data are provided with this paper.

## Code availability

All the scripts employed to analyze the RNA-seq data are accessible through the Github electronic notebook[96] (https://doi.org/10.5281/ZENODO.7992055).

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

## Acknowledgements

We thank Maya Lazar for guidance with RNA-seq data. We thank the Helmholtz-Zentrum Berlin für Materialien und Energie for the allocation of synchrotron radiation beamtime on BAMline, specifically Henning

Markötter and Michael Sintschuk for beamtime setup help. Thanks are extended to Sage Fitzgerlad for her assistance with SEM imaging at UTS. Fieldwork and sequencing costs were funded by the University of Technology Chancellor's Postdoctoral Research Fellowship and Australian Research Council Discovery Early Career Research Award (DE190100142) awarded to E.F.C. The contribution of C.B. and A.D. to the project was funded by a Rolex Award for Enterprise (to E.F.C) . The contribution of F.S., P.Z., and T.M. was funded by the European Research Council (ERC) under the European Union's Horizon 2020 research and innovation program (grant agreement no. 755876 to T.M.).

## Author contributions

E.F.C. and T.M. conceived the project. E.F.C. carried out the sample collection. C.B. completed the RNA extractions. E.F.C., A.D., and T.M. conducted the SEM imaging. E.F.C and A.D. completed the skeletal morphometrics and material structural measurements. F.S., P.Z., and T.M. conducted the skeleton X-Ray imaging. F.S. carried out molecular analyses, reconstructed the tomography data, and analyzed all data. F.S. and E.F.C. wrote the manuscript. All authors contributed to the editing and refinement of the final manuscript.

## Competing interests

The authors declare no competing interests.
