## [Peer Review File · Nature Communications]

The role and risks of selective adaptation in extreme coral habitatsREVIEWER COMMENTS

Reviewer #1 (Remarks to the Author):

The manuscript entitled 'Lessons from extreme – the role and risks of selective adaptations' provides a thorough assessment of genetic and energetic mechanisms together with a detailed evaluation of Porites skeletal characteristics from an extreme site i.e. mangroves, and a nearby reef site. Studies in extreme environments are comparatively lacking yet, as the paper outlines, there has been much interest in these environments recently due to their potential as climate change refugia. But as is often the case, there are biological trade-offs, which this paper clearly demonstrates thereby making the case for caution in relying on these sites for future reef management. Overall, the paper is extremely well written with very few grammatical errors and the figures are beautifully presented. The experimental design has been well thought-out (although I do have some questions on some small aspects of this) and although the sample size is small (which the authors justify well), the results are clear cut. As such, I recommend the paper for publication with some very minor edits/suggestions. Line 186 to 191: I see your point here, but conditions are likely to get harsher in the same direction – hence isn't nature already selecting for the individuals (and their genetics) that are likely to do better anyway?

Line 220-221: not just waves, but also light

Line 264: change to 'under future harsh(er) conditions.'

Line 265: remove 'future' as repeats from line above.

Some repetition in the discussion that can be removed – particularly around the discussion of the importance of loss of genetic diversity and its associated implications for these extreme habitat types treated as climate refugia (e.g. discussed on line ~180 – 192 and again around 250 to 265).

Figure 5: nice well thought out summary of the research outcomes – but probably not really necessary. Would work better as a graphical abstract.

Line 287: Perhaps it would be useful for future researchers to have a more quantitative view of what the abiotic factors are in the mangrove site in question? In figure 5 – you say that light, pH etc are lower at the mangrove site – but if you had a table in supp with some abiotic data comparing between the sites, it would provide more insight into the site differences.

Line 291: Colonies are said to have been collected from a depth of 1 to 1.5 m. Ensuring comparable depths are sampled between locations is important given the nature of this study. Yet the photos shown in fig 1 – show very different depth environments. I assume this is just for demonstration purposes?

Line 292: How far apart were the corals sampled – stated over 5 m to reduce the potential for sampling clonal genotypes, but I think its important to state exactly how far each of the colonies is from each other given that there is a lot of discussion about genetic variability within a site. i.e. are you likely to get more genetic variation when colonies are further apart?

Line 292: Was there consistency in sampling comparable sized colonies as this (age) might also influence the skeletal properties?

Line 292: Where on the colony were the corals sampled? Coral skeletons are known to have morphological differences on top of the colony versus more cryptic locations on the colony. So there needs to be consistency at this level in the sampling.

Line 295: so of the 8 colonies per site – 4 were for molecular and 4 were for skeletal analysis – correct?

Line 370-371: - were there differences in old versus new growth?

Reviewer #2 (Remarks to the Author):

I have reviewed the manuscript by Scucchia et al., investigating the gene expression and skeletal structures of corals inhabiting extreme environments. This study seeks to understand the molecular mechanisms underpinning coral resilience in environments that experience higher than normal environmental change. The manuscript is well written, and for the most part the methods and analysis used are sound. The results are likely of interest to a wide audience and provide novel insights for the

coral molecular ecology field. I do have some doubts though regarding some of the interpretations of the data and believe some more clarity is needed. The authors also state that this data has been made publicly available, but I was unable to access both the scripts and RNAseq data? It is difficult for reviewers to provide a thorough review when the data have not been made available. In addition, the conclusions drawn with such a small sample size of $n=4$ do not seem adequate. E.g., A genotype matrix with only 8 individual samples does not really offer enough cover for population genomics work. It is also worrying when authors do not provide any information regarding the number of reads acquired in the dataset. Is there a reason this information was not provided? Detection of SNPs should have used an appropriate sequencing depth, but it is unclear what that was? I have provided more information regarding this query below. Please also see below for my more specific comments:

Line 72: genome-wide SNPs should be transcriptome-wide SNPs as variant calling was based on RNAseq data not genome-based approach

Line 87: "Clustering based on the proportion of shared alleles" How many they are? Very small n number

Line 108: "940 overrepresented gene ontologies" these are terms. There are only 3 gene ontologies (biological process, cellular component, and molecular functions)"

Line 312: This is confusing. You used a Poly A selection, so there shouldn't be any ribosomal RNA right? Why then was SortMERA used – how many ribosomal RNA did you end up? There shouldn't be any with Poly A.

For GO enrichment analysis, what background was used? what statistical test used at which cut off?

For SNP discovery, read length and sequencing depth should be carefully considered. Paired-end strategy with suitable read length (150pb) and sequencing depth ($> 30-40$ M) is required for accurate variant detection, see (among many others for model organisms) Quagliari et al. 2020. I am surprised that this important information was not disclosed.

Recent coral transcriptome papers trying to detect SNPs have used the appropriate sequencing depth. For example, Voolstra et al. 2021 produced an average of 35 M reads per sample to quantify both expression and adaptation in the coral host.

Line 351: "producing a final genotype matrix of 8 individuals and 22,925 SNPs" both the number of SNPs and samples are not adequate for population genomics work.

Scripts and data need to be publicly available they are currently not. This makes it very difficult for the reviewer to properly review this paper.

Point by point response to reviewers

Reviewer 1

- Line 186 to 191: I see your point here, but conditions are likely to get harsher in the same direction - hence isn't nature already selecting for the individuals (and their genetics) that are likely to do better anyway?

We agree with the reviewer and have provided clarity by revising the text. Mangrove corals exposed to conditions that will intensify as the climate changes, but that they do not currently experience (e.g., disease), could be more susceptible than their reef counterparts due to their smaller allele pool. The revised text now reflects this:

- lines 186-193: *“In fact, variability in gene expression levels represents an advantage that favors adaptation to environmental changes and or/stress conditions³⁴, and thus, mangrove corals may be at a disadvantage in the future as conditions become even more extreme and unpredictable due to global climate change. While mangrove corals have adapted to the present-day extreme environmental conditions, if other stressors that are predicted to intensify, such as disease³⁵, impact the mangrove corals, they could be more at risk due to the smaller pool of alleles and gene expression variations compared to the more genetically heterogeneous reef corals”.*

- Line 220-221: not just waves, but also light

While light could also result in similar effects to the skeletal properties as flow, mean daily light levels have been documented to be similar between the reef and mangrove locations of this study in the winter (Camp et al., 2019), and summer observations support these reports. However, transient changes in light levels can also occur due to turbidity, or higher reflectance of light from the sandy bottom typical of mangrove sites.

We have now added the details about the light conditions at the site of collection in the methods section, and added an additional brief description of the differences in coral morphology linked to water flow:

-lines 245-247: *“Moreover, the microgeometry of the coral skeletal surface appears to regulate networks of currents at the polyps surface, controlling sharing of nutrient, O₂ and metabolites generated by the algal endosymbionts⁵⁶”.*

- lines 281-282: *“Similar mean daily light levels have been previously documented at both Woody Isles and Low Isles during the winter⁸”.*

- Line 264: change to ‘under future harsh(er) conditions.’

Changed:

- line 265: *“under future harsh environmental conditions”.*

- Line 265: remove ‘future’ as repeats from line above.

Removed.

- Some repetition in the discussion that can be removed – particularly around the discussion of the importance of loss of genetic diversity and its associated implications for these extreme habitat types treated as climate refugia (e.g. discussed on line ~180 – 192 and again around 250 to 265).

We have removed repetitions in the discussion paragraphs suggested by the reviewer. The revised paragraphs are now in lines 186-193 (modified as already reported above), and in lines 263-266: “As such, the present superiority in coping with fluctuating stress events of mangrove corals may not guarantee the long-term resilience they have been claimed to possess under future harsh environmental conditions”.

- Figure 5: nice well thought out summary of the research outcomes – but probably not really necessary. Would work better as a graphical abstract.

We thank the reviewer for the suggestion. We will use Figure 5 as a graphical abstract.

- Line 287: Perhaps it would be useful for future researchers to have a more quantitative view of what the abiotic factors are in the mangrove site in question? In figure 5 – you say that light, pH etc are lower at the mangrove site – but if you had a table in supp with some abiotic data comparing between the sites, it would provide more insight into the site differences.

We have added a new supplementary table (Table S3) reporting measurements of abiotic conditions from the literature that were carried out in the study sites of this work and in other mangrove and reef sites around the world.

In the main text:

- line 278: “highly variable abiotic conditions that are characteristic of mangrove systems (Supplemental Table 3)”.

- Line 291: Colonies are said to have been collected from a depth of 1 to 1.5 m. Ensuring comparable depths are sampled between locations is important given the nature of this study. Yet the photos shown in fig 1 – show very different depth environments. I assume this is just for demonstration purposes?

Yes, the pictures shown in Figure 6 are just for demonstration purposes, corals were indeed collected at the same depth (1-1.5 m). We provide here a picture of the exact spot in the mangrove location where one of the colonies was collected.

We chose pictures representing typical reef and mangrove environments, to provide the reader an idea of how these locations generally look. We have now modified the description of Figure 6 to clarify that the pictures are not displaying the exact spots of the sampling, but that they provide a more general view of these sites.

- lines 294-296: “Fig.5: Study sites. Overview of the reef (left image) and mangrove (right image) sites on the Great Barrier Reef adjacent to Low Isles and Woody Isles, respectively, where *P. lutea* corals were collected. Pictures provide a typical view of *P. lutea* found in these two environments”.

- Line 292: How far apart were the corals sampled – stated over 5 m to reduce the potential for sampling clonal genotypes, but I think its important to state exactly how far each of the colonies is from each other given that there is a lot of discussion about genetic variability within a site. i.e. are you likely to get more genetic variation when colonies are further apart?

The pattern of genetic differentiation does not always match the spatial pattern of population sampling, since short-range dispersal can be locally more differentiating than long-range dispersal (Baums et al., 2006). Thus, we aimed to perform sampling to cover the same aerial extent in both sites, focusing on avoiding sampling potential clonal genotypes (colonies were always sampled 5-6 meters apart). Although high clonality between colonies < 5m apart has been observed for other coral species (Baums et al., 2006), such information for *P. lutea* in our study area is currently missing, so future work may focus on investigating clonality levels in both Woody and Low Isles.

We have increased precision stating the exact distance:

-line 284: “5-6 m apart to minimize the potential of sampling clonal genotypes⁵⁷”.

- Line 292: Was there consistency in sampling comparable sized colonies as this (age) might also influence the skeletal properties?

We sampled colonies of comparable size and obtained small colonies with a chisel to account for age based on size. We have increased precision stating the exact size range of 4-6 cm.

- line 283-285: “Colonies (9-10 per site) of comparable size were sampled with a chisel to obtain small colonies (4-6 cm total length) at both sites”.

- Line 292: Where on the colony were the corals sampled? Coral skeletons are known to have morphological differences on top of the colony versus more cryptic locations on the colony. So there needs to be consistency at this level in the sampling.

We collected the entire small colony as noted in the methods, and consistently sampled the skeletal parameters from the top portion of the colony. We have further highlighted this point in the methods, noting that it was standardized across all replicates and both habitats:

- lines 364-366: “All skeletal parameters were consistently measured from the top portion of the colony across all replicates and for both habitats”.

- Line 295: so of the 8 colonies per site – 4 were for molecular and 4 were for skeletal analysis – correct?

We have corrected our inconsistencies in the number of total colonies collected. In total, 9-10 colonies per site were collected and assigned to molecular analysis (4 per site) or skeleton physical property analysis (5-6 per site).

This is now clarified in the methods:

- lines 283-284: “Colonies (9-10 per site) were sampled with a chisel”

- lines 286-288: “Once on the research vessel, colonies were assigned to either molecular analysis (4 per site) or skeleton physical property analysis (5-6 per site)”.

- Line 370-371: - were there differences in old versus new growth?

We found no difference in old versus new growth (see Supplemental Figure 2). We have now clarified this point in the figure description and in the paragraph describing the statistical tests performed in the Methods section:

- lines 383-386: “Statistical significance ($p < 0.05$) of differences between habitats was assessed with parametric unpaired t-test (calyx area, hardness between old and new growth) and non-parametric Mann-Whitney test (hardness – old and new growth analyzed together, porosity and density) using the GraphPad Prism software (v9.0.0)(GraphPad Inc., San Diego, CA, USA)”.

-Supplemental Figure 2 description: “No statistical difference was found in old versus new growth between reef and mangrove corals (unpaired t-test)”.

Reviewer 2

- The authors also state that this data has been made publicly available, but I was unable to access both the scripts and RNAseq data? It is difficult for reviewers to provide a thorough review when the data have not been made available.

We are sorry for the misunderstanding, the reviewer link to see the raw sequencing data deposited in NCBI before publishing the dataset was supposed to be in the manuscript. We have set the raw sequencing data as public, which can now be accessed through <https://www.ncbi.nlm.nih.gov/bioproject/PRJNA912580>. We have added this link in the Data Availability section, see line 411.

The scripts employed to analyze the sequencing data are publicly available in the Github repository https://github.com/fscucchia/Plutea_mangrove_reef, as already mentioned in the Code Availability section (line 413). We have re-checked that the link originally put in the manuscript works and that it is open access.

- In addition, the conclusions drawn with such a small sample size of n=4 do not seem adequate. E.g., A genotype matrix with only 8 individual samples does not really offer enough cover for population genomics work.

Line 351: “producing a final genotype matrix of 8 individuals and 22,925 SNPs” both the number of SNPs and samples are not adequate for population genomics work.

We acknowledge the concern of the reviewer regarding the sample size and the number of SNPs analyzed. Accurate estimation of genetic differentiation in natural populations using limited sample sizes is one of the central issues in population genetic studies. In the literature, there is wide variability in sample size used for population genomic work, indicating that there is a lack of a clear universal sample size rule. This is largely due to the dependence of genetic differentiation dynamics on many factors, including species identity, intrinsic life-history traits, evolutionary and demographic history, strength of natural selection, spatial distribution of the species. Thus, a universal rule-of-thumb for sample size may not be able to sufficiently address these complexities across the entire animal kingdom. Multiple studies have shown that genetic differentiation among populations can be accurately estimated even with small sample numbers (<10) in a variety of taxa, including mammals, fish, insects (Hoban et al., 2013; Nazareno et al., 2017; Phillips et al., 2019; Qu et al., 2020; Trask et al., 2011; Willing et al., 2012). Several works have also demonstrated that increasing the sample size does not improve, or only minorly improves, the accuracy of the genetic diversity estimation (Hoban et al., 2013; Kumasaka et al., 2010; Qu et al., 2020; Zeggini et al., 2005). It also recurrently appears that, even with very small sample size, accurate estimates of genetic diversity can be obtained by screening large numbers of SNPs (>1,500)(Jeffries et al., 2016; Nazareno et al., 2017; Qu et al., 2020; Zimmerman et al., 2020). Typically, population genomic studies on corals (Drury et al., 2016; Hagedorn et al., 2021; Lundgren et al., 2013; Quigley et al., 2020; Smith et al., 2022) and other organisms (Flesch et al., 2018; Gaughran et al., 2018; Hoban et al., 2013; Jeffries et al., 2016;

Kumasaka et al., 2010; Narum et al., 2008; Qu et al., 2020; Williamson et al., 2022; Willing et al., 2012; Zimmerman et al., 2020)(these are just few of the many examples that can be found in the literature) use <30,000 SNPs, most of them using between 1,000 to 21,000 SNPs. The number of SNPs that we employed (22,925) falls within this range.

But we acknowledge using a low sample size and have reformulated our statements regarding the number of samples and SNPs that we employed, to provide a clearer view of the relationship between sample size and SNPs number in the field of population genomic with respect to our estimation of genetic differentiation in *P. lutea*.

- lines 195-204: *“Our small sample size may have only limited power for adequate diagnosis of the genetic variation between populations. Yet, there is wide variability in sample size used for population genomic work, indicating that there is no universal sample size rule allowing to accurately depict the majority of genetic variation across populations and taxa (reviewed in³⁶). However, it recurrently appears that even with restricted sample sizes (< 10) it is possible to adequately differentiate between populations³⁷⁻⁴², especially in the case of populations with confined geographical distributions³⁶, such as for corals living in mangrove lagoons and adjacent reefs. Multiple studies have also demonstrated that increasing the sample size does not improve, or only minorly improves, accuracy of the genetic diversity estimation^{37,41,43,44}. Moreover, in the case of very small sample size, screening large numbers of SNPs (>1,500) can produce accurate estimates of genetic diversity^{40,41,45,46}”.*

- lines 351-353: *“identifying 22,925 SNPs, which fall in the range of other population genomic studies on corals⁷⁸⁻⁸² and other organisms^{37,41,42,44-46,83-86} that employ < 30,000 SNPs to assess genetic differentiation”.*

- It is also worrying when authors do not provide any information regarding the number of reads acquired in the dataset. Is there a reason this information was not provided? Detection of SNPs should have used an appropriate sequencing depth, but it is unclear what that was?

We overlooked this aspect initially, in the interest of brevity, but have now added the information about the total number of reads acquired:

- lines 306-307: *“Approximately 749 million total 150 bp paired-ends reads (47 ± 4 million reads per sample, reported as mean \pm SEM) were sequenced on an on an Illumina NovaSeq 6000”.*

- Line 72: genome-wide SNPs should be transcriptome-wide SNPs as variant calling was based on RNAseq data not genome-based approach.

Corrected, line 74.

- Line 87: *“Clustering based on the proportion of shared alleles”* How many they are? Very small n number.

Please see our previous answer about the number of samples (*n*). Based on 22,925 SNPs, the identity-by-state analysis generated a *n* x *n* matrix of a total of 80,725 genotypes (0,1,2). The matrix was used to cluster the samples according to their genotype similarity. We have now added these details to better explain the clustering.

- lines 356-357: *“A final *n* x *n* matrix of 8 individuals and a total of 80,725 genotypes (0,1,2) was employed for hierarchical clustering”.*

- Line 108: “940 overrepresented gene ontologies” these are terms. There are only 3 gene ontologies (biological process, cellular component, and molecular functions)”.

Corrected, line 110: “*GO terms*”.

- Line 312: This is confusing. You used a Poly A selection, so there shouldn't be any ribosomal RNA right? Why then was SortMeRNA used – how many ribosomal RNA did you end up? There shouldn't be any with Poly A.

Ribosomal RNA is greatly reduced in Poly(A)-selected libraries, but not entirely. To ensure obtaining cleaned reads with the highest quality, we used SortMeRNA and other quality-improving tools, as detailed in the methods, obtaining a quality score above 35 for all samples.

- For GO enrichment analysis, what background was used? what statistical test used at which cut off?

As foreground group, we used all genes analyzed in DESeq2 and, as background, we used all significant differentially expressed genes. We performed the enrichment analysis using the default ‘Wallenius’ model, and identified statistically significant enriched GO terms with a cutoff of $p < 0.05$. We have added this information in the Methods section:

- lines 336-339: “*All genes analyzed in DESeq2 were used as foreground group and all significant differentially expressed genes as background. Enrichment analysis was performed using the default ‘Wallenius’ model, and statistically significant enriched GO terms were selected with a cutoff of $p < 0.05$* ”.

- For SNP discovery, read length and sequencing depth should be carefully considered. Paired-end strategy with suitable read length (150pb) and sequencing depth (> 30-40 M) is required for accurate variant detection, see (among many others for model organisms) Quagliari et al. 2020. I am surprised that this important information was not disclosed. Recent coral transcriptome papers trying to detect SNPs have used the appropriate sequencing depth. For example, Woolstra et al. 2021 produced an average of 35 M reads per sample to quantify both expression and adaptation in the coral host.

Indeed, suitable read length and sequencing depth are critical for accurate SNPs discovery. We have added this information in lines 306-307, as answered above.

- Scripts and data need to be publicly available they are currently not. This makes it very difficult for the reviewer to properly review this paper.

All scripts and data are readily available as mentioned in our answer above.

References

- Baums, I. B., Miller, M. W., & Hellberg, M. E. (2006). geographic variation in clonal structure in a reef-building caribbean coral, *acropora palmata*. *Ecological Monographs*, 76(4), Article 4. [https://doi.org/10.1890/0012-9615\(2006\)076\[0503:GVICSI\]2.0.CO;2](https://doi.org/10.1890/0012-9615(2006)076[0503:GVICSI]2.0.CO;2)
- Camp, E., Edmondson, J., Doheny, A., Rumney, J., Grima, A., Huete, A., & Suggett, D. (2019). Mangrove lagoons of the Great Barrier Reef support coral populations persisting under extreme environmental conditions. *Marine Ecology Progress Series*, 625, 1–14. <https://doi.org/10.3354/meps13073>
- Drury, C., Dale, K. E., Panlilio, J. M., Miller, S. V., Lirman, D., Larson, E. A., Bartels, E., Crawford, D. L., & Oleksiak, M. F. (2016). Genomic variation among populations of threatened coral: *Acropora cervicornis*. *BMC Genomics*, 17(1), 286. <https://doi.org/10.1186/s12864-016-2583-8>
- Flesch, E. P., Rotella, J. J., Thomson, J. M., Graves, T. A., & Garrott, R. A. (2018). Evaluating sample size to estimate genetic management metrics in the genomics era. *Molecular Ecology Resources*, 18(5), 1077–1091. <https://doi.org/10.1111/1755-0998.12898>
- Gaughran, S. J., Quinzin, M. C., Miller, J. M., Garrick, R. C., Edwards, D. L., Russello, M. A., Poulakakis, N., Ciofi, C., Beheregaray, L. B., & Caccone, A. (2018). Theory, practice, and conservation in the age of genomics: The Galápagos giant tortoise as a case study. *Evolutionary Applications*, 11(7), 1084–1093. <https://doi.org/10.1111/eva.12551>
- Hagedorn, M., Page, C. A., O’Neil, K. L., Flores, D. M., Tichy, L., Conn, T., Chamberland, V. F., Lager, C., Zuchowicz, N., Lohr, K., Blackburn, H., Vardi, T., Moore, J., Moore, T., Baums, I. B., Vermeij, M. J. A., & Marhaver, K. L. (2021). Assisted gene flow using cryopreserved sperm in critically endangered coral. *Proceedings of the National Academy of Sciences*, 118(38), e2110559118. <https://doi.org/10.1073/pnas.2110559118>
- Hoban, S., Gaggiotti, O., ConGRESS Consortium, & Bertorelle, G. (2013). Sample Planning Optimization Tool for conservation and population Genetics (SPOTG): A software for choosing the appropriate number of markers and samples. *Methods in Ecology and Evolution*, 4(3), 299–303. <https://doi.org/10.1111/2041-210x.12025>
- Jeffries, D. L., Copp, G. H., Lawson Handley, L., Olsén, K. H., Sayer, C. D., & Hänfling, B. (2016). Comparing RADseq and microsatellites to infer complex phylogeographic patterns, an empirical perspective in the Crucian carp, *Carassius carassius*, L. *Molecular Ecology*, 25(13), 2997–3018. <https://doi.org/10.1111/mec.13613>
- Kumasaka, N., Yamaguchi-Kabata, Y., Takahashi, A., Kubo, M., Nakamura, Y., & Kamatani, N. (2010). Establishment of a standardized system to perform population structure analyses with limited sample size or with different sets of SNP genotypes. *Journal of Human Genetics*, 55(8), 525–533. <https://doi.org/10.1038/jhg.2010.63>
- Lundgren, P., Vera, J. C., Peplow, L., Manel, S., & van Oppen, M. J. (2013). Genotype – environment correlations in corals from the Great Barrier Reef. *BMC Genetics*, 14(1), 9. <https://doi.org/10.1186/1471-2156-14-9>
- Narum, S. R., Banks, M., Beacham, T. D., Bellinger, M. R., Campbell, M. R., Dekoning, J., Elz, A., Guthrieiii, C. M., Kozfkay, C., Miller, K. M., Moran, P., Phillips, R., Seeb, L. W., Smith, C. T., Warheit, K., Young, S. F., & Garza, J. C. (2008). Differentiating salmon populations at broad and fine geographical scales with microsatellites and single nucleotide polymorphisms. *Molecular Ecology*, <https://doi.org/10.1111/j.1365-294X.2008.03851.x>
- Nazareno, A. G., Bemmels, J. B., Dick, C. W., & Lohmann, L. G. (2017). Minimum sample sizes for population genomics: An empirical study from an Amazonian plant species. *Molecular Ecology Resources*, 17(6), 1136–1147. <https://doi.org/10.1111/1755-0998.12654>
- Phillips, J. D., Gillis, D. J., & Hanner, R. H. (2019). Incomplete estimates of genetic diversity within species: Implications for DNA barcoding. *Ecology and Evolution*, 9(5), 2996–3010. <https://doi.org/10.1002/ece3.4757>
- Qu, W., Liang, N., Wu, Z., Zhao, Y., & Chu, D. (2020). Minimum sample sizes for invasion genomics: Empirical investigation in an invasive whitefly. *Ecology and Evolution*, 10(1), 38–49. <https://doi.org/10.1002/ece3.5677>
- Quigley, K. M., Bay, L. K., & Oppen, M. J. H. (2020). Genome-wide SNP analysis reveals an increase in adaptive genetic variation through selective breeding of coral. *Molecular Ecology*, 29(12), 2176–2188. <https://doi.org/10.1111/mec.15482>

- Smith, E. G., Hazzouri, K. M., Choi, J. Y., Delaney, P., Al-Kharafi, M., Howells, E. J., Aranda, M., & Burt, J. A. (2022). Signatures of selection underpinning rapid coral adaptation to the world's warmest reefs. *Science Advances*, 8(2), eabl7287. <https://doi.org/10.1126/sciadv.abl7287>
- Trask, J. A. S., Malhi, R. S., Kanthaswamy, S., Johnson, J., Garnica, W. T., Malladi, V. S., & Smith, D. G. (2011). The effect of SNP discovery method and sample size on estimation of population genetic data for Chinese and Indian rhesus macaques (*Macaca mulatta*). *Primates*, 52(2), 129–138. <https://doi.org/10.1007/s10329-010-0232-4>
- Williamson, J. E., Gillings, M. R., Nevatte, R. J., Harasti, D., Raoult, V., Ghaly, T. M., Stow, A. J., Smith, T. M., & Gaston, T. F. (2022). Genetic differentiation in the threatened soft coral *Dendronephthya australis* in temperate eastern Australia. *Austral Ecology*, 47(4), 804–817. <https://doi.org/10.1111/aec.13160>
- Willing, E.-M., Dreyer, C., & van Oosterhout, C. (2012). Estimates of Genetic Differentiation Measured by F_{ST} Do Not Necessarily Require Large Sample Sizes When Using Many SNP Markers. *PLoS ONE*, 7(8), Article 8. <https://doi.org/10.1371/journal.pone.0042649>
- Zeggini, E., Rayner, W., Morris, A. P., Hattersley, A. T., Walker, M., Hitman, G. A., Deloukas, P., Cardon, L. R., & McCarthy, M. I. (2005). An evaluation of HapMap sample size and tagging SNP performance in large-scale empirical and simulated data sets. *Nature Genetics*, 37(12), 1320–1322. <https://doi.org/10.1038/ng1670>
- Zimmerman, S. J., Aldridge, C. L., & Oyler-McCance, S. J. (2020). An empirical comparison of population genetic analyses using microsatellite and SNP data for a species of conservation concern. *BMC Genomics*, 21(1), 382. <https://doi.org/10.1186/s12864-020-06783-9>

REVIEWERS' COMMENTS

Reviewer #1 (Remarks to the Author):

The authors have addressed my comments thoroughly and have included additional background context data. As such I recommend the paper for publication. I look forward to seeing the final end product.

Reviewer #2 (Remarks to the Author):

I have now read through the revised manuscript by Scucchia et al. and I believe the authors have thoroughly addressed my initial comments and queries. Specifically regarding the bioinformatics, I can now confidently say that the conclusions drawn are appropriate for the data provided and the analysis undertaken. The study provides valuable knowledge regarding the future importance of corals in extreme environments, and is backed up with the correct type of data to support their claims. The results and methods are sound, and now transparent enough for the work to be reproduced, but also for anyone interested to understand how they reached their conclusions. Overall, this is an interesting study that supports novel findings for the field. I do have a few very minor comments, but really very minor and mostly grammatical errors. See my specific comments below:

Line 129 – Should this be $p < 0.001$? In some cases, you give the exact p value and, in some cases, you state the threshold? Similar for line 132, should it be $p < 0.05$? Perhaps either display the exact p value, or state it like you have in line 130 ($p < 0.0001$). Either way it would be nice to keep it consistent throughout the results.

Line 221 – The same sentence appears to have been repeated twice here: “In fact water motion can influence.....” Please remove one.

Line 227 – Should this be ‘multi stressor’ rather than multi stressors? Check this also on line 233.

Line 242 – Check grammar here. Perhaps should be: ‘between mangrove and reef corals’? Rather than ‘mangroves’. (see also line 251).

Line 251 – Please check the sentence here. I think you mean to say: “This implies that it is not only the.....”

Line 263-266 – I think you need to back up this statement with a reference. E.g., make it clear which previous studies have claimed these corals are more resilient to stressors.

Point by point response to Reviewers

Reviewer 2

- Line 129 – Should this be $p < 0.001$? In some cases, you give the exact p value and, in some cases, you state the threshold? Similar for line 132, should it be $p < 0.05$? Perhaps either display the exact p value, or state it like you have in line 130 ($p < 0.0001$). Either way it would be nice to keep it consistent throughout the results.

We have corrected the p values increasing consistency throughout the results:

- line 129: “ $p < 0.01$ ”

- line 132: “ $p < 0.05$ ”

- Line 221 – The same sentence appears to have been repeated twice here: “In fact water motion can influence.....” Please remove one.

We have removed the repetition:

-line 222: “mangrove lagoons are not subjected to strong wave action and currents, thus the skeleton morphotype of mangrove corals may be the product of local adaptation to a more protected location. *In fact, water motion can influence the skeleton development dynamics of stony corals, with lower flow intensity leading to the formation of less compact and reduced thickness*”.

- Line 227 – Should this be ‘multi stressor’ rather than multi stressors? Check this also on line 233.

Corrected.

- Line 242 – Check grammar here. Perhaps should be: ‘between mangrove and reef corals’? Rather than ‘mangroves’. (see also line 251).

Corrected.

- Line 251 – Please check the sentence here. I think you mean to say: “This implies that it is not only the.....”

Corrected.

- Line 263-266 – I think you need to back up this statement with a reference. E.g., make it clear which previous studies have claimed these corals are more resilient to stressors.

We have added references to back up our statement:

-line 265: “the long-term resilience they have been claimed to possess^{6,11}”.